# Bat-human interactions and associated factors among communities in Bundibugyo District, Uganda: A cross-sectional study

**James Natweta Baguma**[1,2]*, **Shamilah Namusisi**[3], **Lesley Rose Ninsiima**[1,2], **Rogers Musiitwa**[1,2], **Bridget Nagawa Tamale**[1], **Mathius Amperiize**[1], **Douglas Bulafu**[1], **James Muleme**[1,2], **David Musoke**[1], **Clovice Kankya**[2], **Charles Drago Kato**[2,3]

1 Department of Disease Control and Environmental Health, School of Public Health, College of Health Sciences, Makerere University, Makerere Hill, Kampala, Uganda, 2 Department of Biosecurity, Ecosystems and Veterinary Public Health, School of Biosecurity, Biotechnology and Laboratory Sciences, College of Veterinary Medicine, Animal Resources and Biosecurity, Makerere University, Makerere Hill, Kampala, Uganda, 3 Africa One Health University Network (AFROHUN), Kampala, Uganda

* bagumajamesnat@gmail.com

## Abstract

More than 70% of new, emerging, and reemerging infectious diseases are from animal origin. Human interaction with bats has been associated as a driver for various fetal zoonoses, including numerous viral diseases of bat-origin. A lot of serological evidence has been gathered around human-bat interaction, yet very little is known regarding the underlying risk factors at community level. This study was aimed at understanding the human-bat interactions and associated factors among communities in Bundibugyo District in Uganda. A cross-sectional study was conducted using both qualitative and quantitative data collection methods in Harugale, Burondo and Ntandi Subcounties in Bundibugyo District between November 2022 and March 2023. A total of 344 participants were interviewed using a structured questionnaire. Key Informant Interviews (KIIs) were also conducted among purposively selected individuals who have vast knowledge on human-bat interaction. Proportional piling and FGDs were conducted among groups of men, women, and youths to get their insights into human-bat interaction. The study revealed that 54.1% of the respondents were males, 42.1% were aged above 40 years. Households headed by males (APR = 1.31, 95% CI:1.07-1.62, Batwa communities (APR = 3.03, 95% CI:1.87-3.94), residing in urban areas (APR = 1.72, 95 CI%:1.35-2.20), trading of food and animal products (APR = 0.6, 95 CI%:0.36-0.99), no occupation (APR = 0.27, 95 CI%:0.12-0.57) and residing in incomplete houses (APR = 1.57, 95 CI%:1.25-1.98) were significantly associated with exposure of humans to bats. There was high exposure of humans to bats in Ntandi compared to Burondo and Harugale. Women groups highlighted use of bat repellants and killing of bats using sticks as the measures to reduce human bat interaction during the focus group discussions (FGDs). Generally, there

**Data availability statement:** All relevant data are within the manuscript and the Supporting Information files.

**Funding:** This study was under the Strategies to Prevent Spillover Project implemented by the Africa One Health University Network (AFROHUN) funded by United States Agency for International Development (USAID). The funders had no role in study design, data collection and analysis, decision to publish, or preparation of the manuscript.

**Competing interests:** The authors have declared that no competing interests exist.

is high exposure to bats among human communities in Bundibugyo district which increases the risk of zoonotic disease transmission at human-bat interface. Findings from this study aim to enable the one health interventions to reduce bat-human interaction potential risks in both urban and rural areas and support design of feasible interventions for Bundibugyo district and Uganda at large.

## Introduction

Globally, more than 70 percent of new, emerging, and re-emerging infectious diseases originate from animals [1]. The COVID-19 pandemic and the recent Ebola Outbreak in Uganda demonstrates the tremendous risks that can occur when zoonotic diseases spill over to humans nationally and internationally [2].

The West Africa epidemic of Ebola virus disease (EVD) started in Guinea in December 2013 and resulted in a total of 28,646 reported cases and 11,323 deaths [3]. Most transmissions happened in Guinea, Sierra Leone, and Liberia; however, the outbreak spread out to other seven states.

Over the past four decades there has been an increase in the rate at which zoonotic diseases occur in humans. Interaction between humans and animals has been implicated as a primary risk factor for respective high impact zoonoses, including many bat-origin viral diseases [4]. The consequences of such outbreaks have often been overwhelming, threatening the already fragile economic and healthcare systems of affected countries [5]. Bats (Order Chiroptera) are the known host reservoirs for a range of viruses and their role in recent emergent infectious diseases has been firmly established [6]. More than 60 viruses have been detected in bat tissue with bats implicated as reservoir hosts for highly pathogenic Nipah, Hendra, Lyssa and Ebola viruses [7].

Despite the paucity of information available on interactions between humans and bats, bats are suspected to be involved in zoonotic transmissions. For example, it was revealed that the index case of the outbreak in 2007 in Luebo, Democratic Republic of Congo (DRC) was asymptomatically infected through contact with bat bushmeat [8]. In 2013, there was another one in West Africa where the index case was a 2-year-old boy who was suspected to have played with a roost of insectivorous bats inside a hollow tree. But for most outbreaks, the initial source of zoonotic transmission has not been identified [9]. With the overwhelming human-bat interactions that increase risks of transmissions, activities such as hunting, butchering, and consuming bats are frequent and diverse in tropical Africa. These practices can potentially lead to transmission of zoonotic pathogens through animal bites, scratches, contact with infected body fluids, tissues and excrements that can also contaminate fruits [10].

The consumption of bats by humans has been linked to population depletions, the transmission of zoonotic diseases and the loss of ecosystem services such as pollination and forest regeneration [11]. Among zoonotic viruses for which bats serve as reservoirs are the henipavirus, lyssavirus, filovirus, Ebola virus and rabies. These zoonotic pathogens are potentially transmitted through animal bites, scratches, body

fluids, tissues, and excrement. It is important to note that colonies of bats located in urban areas raise additional concerns about zoonotic diseases transmissions.

In Bundibugyo district in Southwestern Uganda, interactions among humans, wildlife, and livestock are likely to be common. This is attributed to disruption of natural habitats initially belonging to bats by humans forcing them into close proximity with humans [11]. A study conducted in West Africa hypothesized close contact between humans and wildlife at the human, animal and environment interface to be a risk factor in the emergence of zoonotic infectious diseases [12]. However, information on bat hunting and consumption practices, as well as indirect contacts, is scarce. Understanding of the risk factors, knowledge, awareness, and perceptions of communities towards bat-related diseases is also limited.

Therefore, this study was conducted to determine the drivers of contact between human populations and bats, awareness, beliefs, and perceptions regarding bats in Bundibugyo district in Southwestern Uganda, a region considered at risk of zoonotic outbreaks.

## Materials and methods

### Study place, population and design

The study adopted a cross-sectional study design using both qualitative and quantitative methods. It was carried out between 2nd November 2022 and 25th March 2023 among selected household heads and community members from Ntandi, Harugale and Burondo sub counties in Bundibugyo district. It is located approximately 356 kilometers (221 miles) west of Kampala. According to recent Uganda Bureau of Statistics (UBOS) estimates, the district has a population of 263,800 people with over 72% living in rural areas. Bundibugyo district boarders the Democratic Republic of Congo and Ntoroko District [13]. It is in proximity with Semuliki National Park, Rwenzori National Park, and Kibaale National Park. The study was conducted in three sub counties that is Burondo Sub County which neighbors Semliki National Park, Harugale Subcounty below which neighbors Rwenzori National Park and Ntandi Town Council which also neighbors Semuliki National Park and lies between Harugale and Burondo as in Fig 1 below. This has led to a high level of human-wildlife-forest ecosystem interactions. Additionally, the predominant ethnicities include the Bamba-Babwisi and the Bakonjo. However, they co-exist with other ethnicities such as the Batooro. The area has plenty of rock shelters and caves that are habitats for wildlife, including bats. The people of Bundibugyo are principally farmers who grow coffee, vanilla, and cocoa to get income. These are facilitated by the two rainy seasons of March to May and July to November [14].

The households were the study unit for quantitative data and any person between 18–65 years of age and has voluntarily accepted to participating in the study and resides in that household was the research unit. The qualitative data involved conducting KIIs (KIIs) to get their expert opinion on human-bat interaction and FGDs with men, women, and youths' groups. A questionnaire was administered to household heads to determine the drivers of human-bat interaction at household level, risk of human exposure to bats, awareness on bats, mitigation measures, profiled dangers and benefits of bats and perceptions. This guided the drafting of the FGD and KII guides to unfold the underlying factors affecting human-bat interaction that had not come out clearly during the surveys such as cultural beliefs on bats. This was further supplemented by participatory epidemiology tools such as proportional piling to further examine the risks related to human interaction with bats. Participants were asked to count beans and distribute counters to show the crops that are destroyed by bats. Once they were happy with the distribution of the counters, the results were recorded as shown in Fig 2.

### Study area

The study was conducted among communities in the three selected sub counties of Bundibugyo district. These included household heads, women, men and youths' groups, key informants such as the District Veterinary Officer, the District Health Officer, the bat hunters, traditional health practitioners, religious leaders, bat consumers, and the sub county leaders among others.

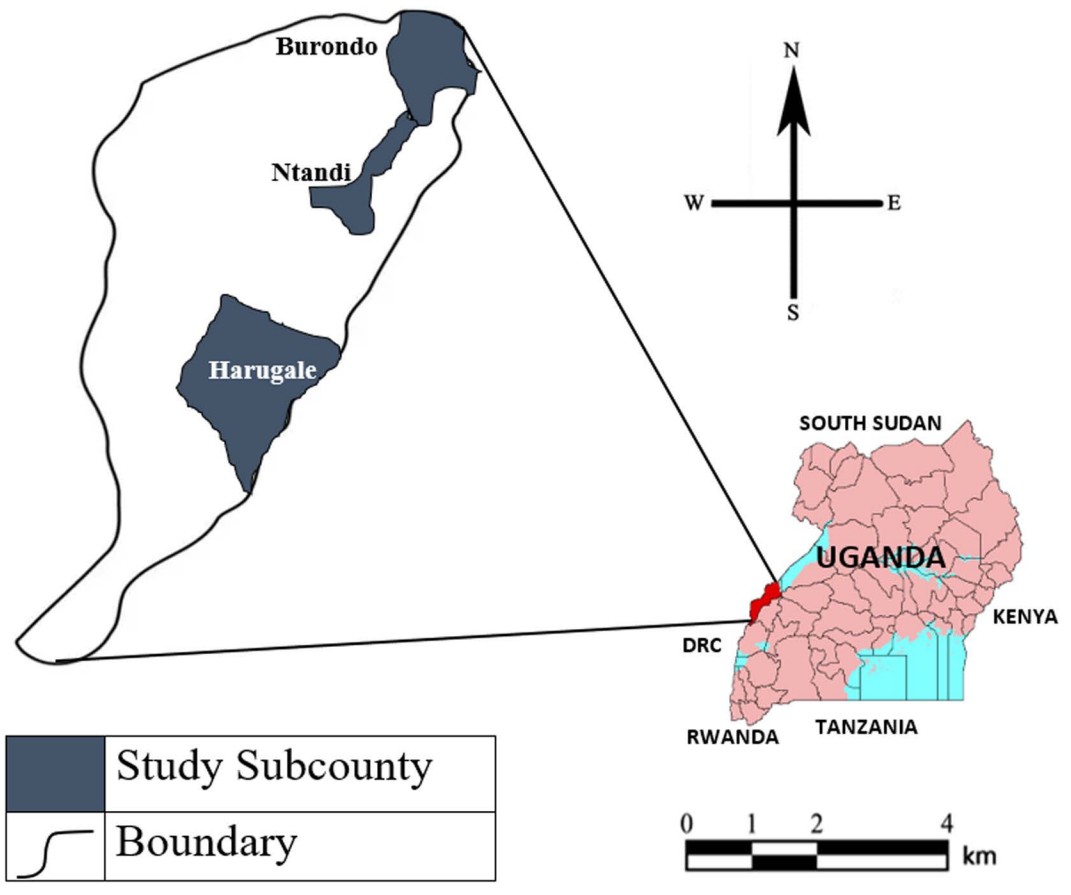

**Fig 1. Map of Bundibugyo showing Ntandi, Harugale and Burondo. PlaniGlobe, http://www.planiglobe.com, CC BY 2.0.**

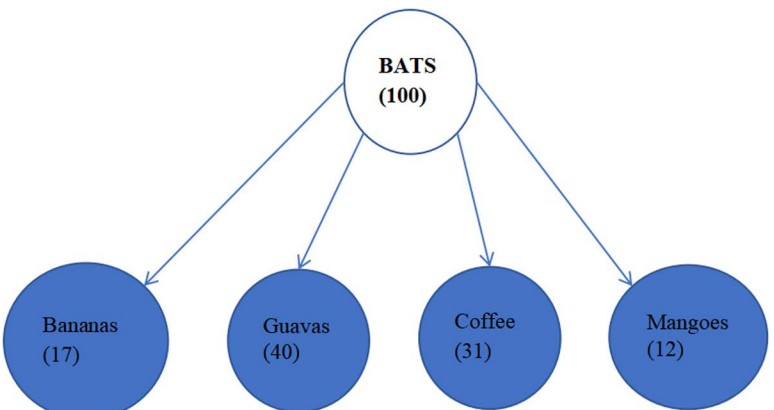

**Fig 2. Proportional piling showing that guavas are the most destroyed by bats.**

The sampling strategy was guided by the Strategies to Prevent Spillover (STOPs) bat monitoring program which generated a map showing several bat roosts including households with bats at that time (https://stopspillover.org/sites/default/files/202403/updated_maps_for_Bundibugyo_District_Uganda-508.pdf). Harugale, Ntandi, and Burondo were purposively selected due to their proximity to bat roosts and wildlife reserves. Households that were included in the study were then randomly selected from the map. Each household on the map was assigned a unique identifier to ensure clear identification. A random number generator was then employed using Microsoft Excel to randomly select households from the list, giving each an equal chance of being included in the study. This was done until we reached our calculated sample size with equal representation and unbiased selection of households. Key informants and FGD participants were selected purposively given their knowledge and expertise as well as recent direct or indirect engagement in activities such as eating bats, hunting, bat guano collection among others.

The sample size was determined using the Kish Leslie [15] formula considering a conservative prevalence of 66% prevalence human-bat interaction [16]. A sampling error of 5% was considered giving a sample size of 344 participants.

Research Assistants (RAs) experienced in both quantitative and qualitative data collection, fluent in English and local dialects spoken by the target respondents were recruited and subjected to a minimum of 4 days training and one day of piloting the tools for reliability and coherence test. Quantitative data was collected using a structured questionnaire installed on the Kobo Collect Software, cleaned using Microsoft Excel and exported to Stata 15 for analysis.

All data collected qualitatively was audio recorded using encrypted recorders. At the end of each day, RAs were required to backed up in a password-protected folder. At the end of data collection, the audio recordings and transcripts were availed for analysis. Our study was limited by response and recall bias during data collection.

## Statistical analysis

Both bivariate and multivariable analyses were conducted to ascertain significant variables. In the bivariate analysis, a significance level of $p < 0.2$ was used, while in the multivariable analysis, the significance level was set at $p < 0.05$. The rationale for using a p-value cutoff of 0.2 in the bivariate analysis was to include potentially relevant variables that may have a weaker association with the outcome but could still be important in the context of the study. This approach allowed for a more comprehensive exploration of factors before applying more stringent criteria in the multivariable analysis. In addition, the independent variables deemed essential to the research question, were kept in the final model because they had potential to influence exposure to bats. Since the outcome (exposure to bats) had a prevalence that exceeded 10%, a modified Poisson regression analysis with robust standard errors was used to report prevalence ratios. Confounding was assessed on all variables that dropped out of the model using stepwise elimination, with assessment prioritized from the most significant to the least significant variable that dropped out.

To measure the level of human-bat exposure, a set of 8 questions were designed to assess the level of exposure of humans to bats among the respondents as used in a previous study conducted in Mindanao, Philippines [17]. A composite score was generated based on summation of all the correct response in relation to whether the respondents are exposed to bats or not. The median score for exposure to bats served as the cutoff point. Participants scoring at or above the median were classified as having high exposure to bats, while those below were classified as having low exposure to bats. Additionally, the median was used as a cutoff because it represented the midpoint of the data distribution, ensuring an equal split between participants with higher and lower levels of exposure to bats.

The questions are listed below.

a. *Do you have bats in your household? Yes/No*

b. *Do you see bats on trees within your compound? Yes/No*

c. *Are there bats at your farm? Yes/No*

d. *Do members in your household hunt bats? Yes/No*

e. *Do children touch and or play with bats? Yes/No*

f. *Have you or your household members found bat faeces in your household? Yes/No*

g. *Have you ever eaten bat meat? Yes/No*

h. *Have any of your friends or relatives close to your household eat bats? Yes/No*

## Qualitative analysis

All audio recorded discussions were then transcribed in English during and after data collection. It was intended to be highly iterative allowing reflections as data is being collected. Through this process, emerging themes were identified and followed up while teams were still in the field to ensure that a particular issue is comprehensively studied and understood. Interviewers were trained in how to identify new information and make follow-up during subsequent interviews. During daily briefings with the team, emerging issues were discussed with the team to identify follow-up issues. This was intended to identify new insights that may not have been anticipated at the design of the study, but which are critical in understanding human-bat interactions and their risks among communities in Bundibugyo district.

Transcribed data were stored as word document and exported into Atlas. ti (Version 8.0) for further management. A code frame was developed and discussed between the team members. The initial code frame was based on the thematic areas developed from the objectives of the study. After the initial code frame has been developed, a painstaking process of reviewing each transcript was conducted to identify new issues but also to establish patterns and analytical reflections for further analysis. After this process, a more focused analysis was undertaken to identify areas of convergence and divergence and identify areas and processes for strengthening the interpretation and explanation of the quantitative results findings.

## Ethical considerations

The ethical approval to conduct the study was sought and obtained from the Makerere University School of Public Health -MAKSPH HDREC990 and permission to conduct the study was obtained from the Uganda National Council for Science and Technology-HS2428ES. This study involved talking to household members in these communities including those that have had negative experiences of bats and those whose psychosocial health might have been affected which might evoke emotional stress, feelings of hopelessness and cause unnecessary distress. To ensure that the study doesn't evoke and cause harm, several measures were proposed to ensure that any ethical issues that may arise are addressed. Informed consent was obtained from all adult participants. Informed Consent was obtained after explaining thoroughly the purpose of the study. Bearing in mind the COVID-19 and Ebola prevention measures where transmission is person to person, we sought informed oral consent for all respondents.

## Results

### Socio-demographic characteristics of the respondents

Three hundred forty-four (344) respondents were included in the study from three sub counties in Bundibugyo district. More than half were males, 186/344 (54.1%) and 158/344 (45.9%) were females. Majority of the respondents were Anglican, 139/344 (40.4%) and majority of the Anglicans were residing in Ntandi 64/139 (64%). The study revealed that none of the respondents in Ntandi Subcounty was Pentecostal 0/55 (0.0%). More than half of the respondents 182/344 (52.9%) had only attended primary school as the highest level of education and only 26/344 (7.5%) had attained tertiary level education. 65/344 (18.9%) of the respondents had not attended school and almost half of these were from Burondo

subcounty 27/65 (41.5%). Most of the respondents were residing in rural areas of Bundibugyo district 215/344 (62.5%). Notably, all the respondents from Ntandi Town Council affirmed that they were residing in the urban areas 129/129 (100%) and the respondents from Burondo 113/215 (52.6%) and Harugale 102/215 (47.4%) confirmed were from the rural areas. The study revealed that more than half of the houses where the respondents were residing were semi-permanent 209/344 (60.8%). The study revealed that crop production was the most cited as the leading source of livelihood income among the various households. More than half of the respondents, 205/344 (59.6%) were earning from crop production especially in cocoa and vanilla. Crop production was more identified in Burondo sub county 95/205 (44.9%) compared to the Ntandi and Harugale. Hunting was more prominent in Burondo Sub County 17/27 (63.3%) where the Batwa community that is well known for hunting resides. Other livelihood activities included hunting 27/344 (7.8%), trading (business) 21/344 (6.1%), Self-employment 21/344 (6.1%) and public service 15/344 (4.4%) among others as shown in Table 1 below. During the FGDs with the various community members, it came out clearly that the most relied on source of income livelihood was crop farming followed by other activities such as trading, hunting, fishing, livestock farming, quarrying, brick laying among others.

Table 1. Socio-demographic factors in Ntandi, Burondo and Harugale sub counties.

| Variables | Category | Burondo n (%) | Harugale n (%) | Ntandi n (%) | Total |
|---|---|---|---|---|---|
| **Sex** | Female | 62 (39.2) | 43 (27.2) | 53 (33.5) | 158 (45.9) |
| | Male | 51 (27.4) | 59 (31.7) | 76 (40.8) | 186 (54.1) |
| **Religion** | Anglican | 45 (32.4) | 30 (21.6) | 64 (46) | 139 (40.4) |
| | Muslim | 8 (30.8) | 0 (0.0) | 18 (69.2) | 26 (7.6) |
| | Pentecostal | 15 (27.3) | 40 (72.7) | 0 (0.0) | 55 (16) |
| | Roman catholic | 15 (68.2) | 6 (27.3) | 1 (4.5) | 21 (6.1) |
| | Seventh-day Adventist | 30 (29.4) | 26 (25.5) | 46 (45.1) | 102 (29.7) |
| **Education level** | Never attended school | 27 (41.5) | 12 (18.5) | 26 (40) | 65 (18.9) |
| | Primary school | 77 (42.3) | 42 (23.1) | 63 (34.6) | 182 (52.9) |
| | Secondary school | 6 (8.1) | 32 (43.2) | 36 (48.6) | 74 (21.5) |
| | Tertiary level | 3 (11.5) | 19 (73.1) | 4 (15.4) | 26 (7.5) |
| **Residential status** | Rural | 113 (52.6) | 102 (47.4) | 0 (0.0) | 215 (62.5) |
| | Urban | 0 (0.0) | 0 (0.0) | 129 (100) | 129 (37.5) |
| **Building structure** | Permanent | 57 (59.4) | 21 (21.9) | 18 (18.8) | 96 (27.9) |
| | Semi-permanent | 55 (26.3) | 60 (28.7) | 94 (45) | 209 (60.8) |
| | Temporary | 1 (2.6) | 21 (53.8) | 17 (43.6) | 39 (11.3) |
| **Floor type** | Cemented | 53 (35.1) | 50 (33.1) | 48 (31.8) | 151 (43.9) |
| | Dusty | 52 (28.1) | 52 (28.1) | 81 (43.8) | 185 (53.8) |
| | Tiled | 8 (100) | 0 (0.0) | 0 (0.0) | 8 (2.3) |
| **Roofing** | With ceiling | 48 (56.5) | 4 (4.7) | 33 (38.8) | 85 (24.7) |
| | Without ceiling | 65 (25.1) | 98 (37.8) | 96 (37.1) | 259 (75.3) |
| **Livelihood activity** | Crop production | 92 (44.9) | 47 (22.9) | 66 (32.2) | 205 (59.6) |
| | Government employee | 3 (20.0) | 8 (53.3) | 4 (26.7) | 15 (4.4) |
| | Housewife | 1 (14.3) | 6 (85.7) | 0 (0) | 7 (2.0) |
| | Hunting | 17 (63.3) | 3 (11.1) | 7 (25.9) | 27 (7.8) |
| | Market (food and animal trade) | 6 (28.6) | 6 (28.6) | 9 (42.9) | 21 (6.1) |
| | No occupation/ unemployed | 1 (2.8) | 14 (38.9) | 21 (58.3) | 36 (10.5) |
| | Self employed | 3 (14.3) | 14 (66.7) | 4 (19.0) | 21 (6.1) |
| | Student | 0 (0.0) | 4 (33.3) | 8 (66.7) | 12 (3.5) |

*"We are lucky that our soil is fertile, and we have rain throughout the year. This enables us to grow our crops and when we harvest, we get money to pay school fees for our children and provide for other basic needs. We grow Cocoa, vanilla, beans, maize, and many other crops. We also rear some goats but those are not so many."* - **(FGD with Women in Burondo)**

## Human bat interaction

The level of exposure of humans to bats was measured with a set of 8 questions as in Table 2 below. The study revealed that more than half of the respondents 179/344 (52.0%) had high exposure to bats and 165/344 (48.0%) of the respondents had low exposure to bats.

The study revealed that more than half of the respondents had been exposed to bats through bat guano in the household 175/344 (50.9%), 60/344 (17.4%) of the respondents had eaten bat meat and majority of these were from Ntandi 35/60 (58.3%). Similarly, an equal percentage of the respondents confirmed that their friends and relatives were also eating bats. It was noted that in 136/344 (39.5%) of the households that were visited, the children were directly in contact with bats through playing and touching them. Almost three-quarters 255/344 (74.1%) of the respondents affirmed that they have bats present in their households and majority of the households that have bats are in Ntandi 107/255 (74.1%). Generally, more than half of the households were exposed to bats with Ntandi having the highest exposure 82/179 (45.8%) as shown in Table 2 below.

Respondents during one of the KIIs highlighted that human-bat interaction varies with the seasons when bats are present and absent. The respondents highlighted that there is usually high exposure to bats in the month of March and low exposure in the month of June and this attributed to the rainy seasons as indicated below.

*"These bats are funny creatures, when there is rain especially in the month of March, they leave their residences and come to our homes, we see so many of them in out compounds and communities, and when the sun is too much like*

**Table 2. Human-bat interaction among communities in Bundibugyo district.**

| Variables | Category | Burondo n (%) | Harugale n (%) | Ntandi n (%) | Total n (%) |
|---|---|---|---|---|---|
| **Friends/Relatives consume bats** | No | 103 (36.3) | 87 (30.6) | 94 (33.1) | 284 (82.6) |
| | Yes | 10 (16.7) | 15 (25.0) | 35 (58.3) | 60 (17.4) |
| **Ever eaten bat meat** | No | 96 (33.8) | 94 (33.1) | 94 (33.1) | 284 (82.6) |
| | Yes | 17 (28.3) | 8 (13.3) | 35 (58.3) | 60 (17.4) |
| **Bat guano in the household** | No | 72 (42.6) | 52 (30.8) | 45 (26.6) | 169 (49.1) |
| | Yes | 41 (23.4) | 50 (28.6) | 84 (48.0) | 175 (50.9) |
| **Children touch and play with bats** | No | 78 (37.5) | 61 (29.3) | 69 (33.2) | 208 (60.5) |
| | Yes | 35 (25.7) | 41 (30.1) | 60 (44.1) | 136 (39.5) |
| **Members in household hunt bats** | No | 106 (33.4) | 99 (31.2) | 112 (35.3) | 317 (92.5) |
| | Yes | 7 (25.9) | 3 (11.1) | 17 (63.0) | 27 (7.8) |
| **Bats in the household farm** | No | 75 (34.1) | 73 (33.2) | 72 (32.7) | 220 (64.0) |
| | Yes | 38 (30.6) | 29 (23.4) | 57 (46.0) | 124 (36.0) |
| **Bats in the trees in the compound** | No | 93 (36.8) | 59 (23.3) | 101 (39.9) | 253 (73.5) |
| | Yes | 20 (22.0) | 43 (47.3) | 28 (30.8) | 91 (26.5) |
| **Bats present in the households** | No | 56 (62.9) | 11 (12.4) | 22 (24.7) | 89 (25.9) |
| | Yes | 57 (22.4) | 91 (35.7) | 107 (42.0) | 255 (74.1) |
| **Level of Exposure to bats-Median** | Low | 74 (44.8) | 44 (26.7) | 47 (28.5) | 165 (48.0) |
| | High | 39 (21.8) | 58 (32.4) | 82 (45.8) | 179 (52.0) |

*in the month of June, you can even spend a day without seeing a bat in the neighborhood."* -**Key Informant, Ntandi Town Council**

## Factors associated with human-bat interaction among communities in Bundibugyo district

At bivariate analysis, households headed by males (CPR = 1.42, 95% CI: 1.14-1.76) had 42% higher prevalence of high exposure to bats compared to households headed by females. Households headed by a roman catholic (CPR = 1.50, 95% CI: 1.09-2.08) had a 50% higher prevalence of high exposure to bats compared to those headed by Anglicans. The prevalence of high exposure to bats was 127% higher in the Batwa communities (CPR = 2.27, 95% CI: 1.49-3.23) compared to the Bakonjo. Households in urban areas (CPR = 1.41, 95% CI: 1.164-1.72) had a 41% higher prevalence rate of high exposure to bats compared to households in rural areas. Similarly, respondents who were living in incomplete houses (CPR = 1.41, 95% CI: 1.15-1.73) had a 41% higher prevalence of high exposure to bats compared to those who were living in complete houses. Households that had no source of income (CPR = 0.28, 95% CI: 0.13-0.59) had a 72% lower prevalence of high exposure to bats compared to the households that were earning from crop production. Households that had storage rooms (CPR = 1.40, 95% CI: 1.12-1.76) had a 40% higher prevalence for high exposure to bats compared to the households that did not have storage rooms.

After adjusting for confounders at multivariable analysis, households headed by males, the Batwa tribe, households in urban areas, households depending on trade in food and animals and no occupation or unemployment were significantly associated with exposure to bats.

Households headed by males were associated with a 31% higher prevalence of high exposure to bats (APR = 1.31, 95% CI:1.07-1.62) compared to those headed by females. The Batwa communities APR = 3.03, 95% CI:1.87-3.94) had a 203% higher prevalence of high bat exposure compared to the Bakonjo. Household in rural areas (APR = 1.72, 95 CI%:1.35-2.20) had a 72% higher prevalence of high bat exposure compared to those in rural areas.

Households engaged in trading of food and animal products (APR = 0.6, 95 CI%:0.36-0.99) had a 40% lower prevalence of high exposure to bats compared to the households who were depending on crop production. Households with no occupation or no employment (APR = 0.27, 95 CI%:0.12-0.57) had a 73% lower prevalence of high exposure to bats compared to those depending on crop production. Residing in incomplete structures (APR = 1.57, 95 CI%:1.25-1.98) was associated with a 57% higher prevalence of exposure to bats compared to residing in complete houses as shown in Table 3 below.

## Awareness on human-bat interaction

Generally, awareness on human-bat interaction varied in Ntandi, Harugale and Burondo sub counties. More than half of the respondents in Ntandi 79/157 (50.3%) were aware that bats enter the house through the roof. Similarly, more than half of the respondents from Ntandi 67/118 (57.3%) were aware that bats carry pathogens that cause diseases such as Marburg. The study revealed that there was high awareness of bats presence in schools in Burondo, Harugale and Ntandi 99/317 (31.2%), 89/317 (28.1%), 129/317 (40.7%) respectively. However, there was low awareness of bats in places of worship as majority of the respondents were not aware especially in Ntandi 109/266 (77.3%). There was also low awareness on trade in bat meat in Burondo, Ntandi and Harugale 21/37 (56.8%), 6/37 (16.2%), 10/37 (27.0%) respectively. Majority of the respondents 122/344 (35.5%) had heard or seen information about bats specifically regarding the diseases associated with human-bat interaction. A radio 99/344 (28.8%) was mentioned as the media where majority of the respondents had heard the information. Similarly, a radio 113/344 (33.9%) was mentioned as the most trusted source of health information about bats. Almost three quarters 250/344 (72.7%) of the respondents confirmed that the information that they come across about bats is not clear and almost all the respondents 324/344 (94.2%) said that there is need for more information regarding human bat interaction most especially how to reduce human-bat interaction 276/344 (80.2%) as shown in Table 4 below.

**Table 3. Factors associated with human-bat interaction among communities in Bundibugyo district.**

| Variable | Attribute | Exposure to bats | | Crude PR (95% CI) | P-values | Adjusted PR (95% CI) | P-values |
|---|---|---|---|---|---|---|---|
| | | Low (n = 165) | High (n = 179) | | | | |
| **Sex of the household head** | Female | 91 (57.6) | 67 (42.4) | 1 | | 1 | |
| | Male | 74 (39.8) | 112 (60.2) | 1.42 (1.14-1.76) | **0.001** | 1.31 (1.07-1.62) | **0.008** |
| **Age of household head** | Below 30 | 39 (49.4) | 40 (50.6) | 1 | | 1 | |
| | 30-40 | 58 (48.3) | 62 (51.7) | 1.02 (0.77-1.35) | 0.887 | 0.96 (0.76-1.21) | 0.737 |
| | Above 40 | 68 (46.9) | 77 (51.3) | 1.04 (0.80-1.37) | 0.726 | 1.08 (0.85-1.36) | 0.542 |
| **Religion of the household head** | Anglican | 73 (52.5) | 66 (47.5) | 1 | | 1 | |
| | Muslim | 13 (48.1) | 14 (51.9) | 1.09 (0.72-1.64) | 0.669 | 0.87 (0.57-1.34) | 0.544 |
| | Pentecostal | 30 (54.5) | 25 (45.5) | 0.96 (0.68-1.34) | 0.801 | 1.17 (0.77-1.78) | 0.464 |
| | Roman catholic | 6 (28.6) | 15 (71.4) | 1.50 (1.09-2.08) | **0.013** | 1.45 (0.80-2.64) | 0.219 |
| | Seventh-day Adventist | 43 (42.2) | 59 (57.8) | 1.22 (0.96-1.55) | **0.109** | 0.97 (0.71-1.31) | 0.840 |
| **Tribe/Ethnicity** | Bakonjo | 39 (60.0) | 26 (40.0) | 1 | | 1 | |
| | Bamba-Babwisi | 113 (47.1) | 127 (52.9) | 1.32 (0.96-1.82) | **0.088** | 1.13 (0.81-1.57) | 0.464 |
| | Batwa | 1 (9.1) | 10 (90.9) | 2.27 (1.59-3.23) | **<0.001** | 3.03 (1.87-4.94) | **<0.001** |
| | Batooro | 12 (42.9) | 16 (57.1) | 1.43 (0.92-2.21) | **0.111** | 1.07 (0.68-1.70) | 0.771 |
| **Period of stay in study area** | No | 5 (71.4) | 2 (28.6) | 1 | | | |
| | Yes | 160 (47.5) | 177 (52.5) | 1.84(0.57-5.97) | 0.311 | | |
| **Level of education** | Never attended school | 37 (56.9) | 28 (43.1) | 1 | | | |
| | Primary school | 88 (48.4) | 94 (51.6) | 1.20 (0.88-1.64) | **0.256** | | |
| | Secondary school | 35 (41.2) | 50 (58.8) | 1.37 (0.98-1.90) | 0.666 | | |
| | Tertiary level | 5 (41.7) | 7 (58.3) | 1.35 (0.78-2.36) | **0.284** | | |
| **Residential status** | Rural | 118 (54.9) | 97 (45.1) | 1 | | 1 | |
| | Urban | 47 (36.4) | 82 (63.6) | 1.41 (1.16-1.72) | **0.001** | 1.72 (1.35-2.20) | **<0.001** |
| **Floor type** | Cemented | 73 (48.3) | 78 (51.7) | 1 | | | |
| | Dusty | 88 (47.6) | 97 (52.4) | 1.01 (0.83-1.25) | 0.888 | | |
| | Tiled | 4 (50.0) | 4 (50.0) | 0.97 (0.48-1.97) | 0.928 | | |
| **Roofing** | With ceiling | 37 (43.5) | 48 (56.5) | 1 | | | |
| | Without ceiling | 128 (49.4) | 131 (50.6) | 0.89 (0.72-1.12) | 0.332 | | |
| **Status of house** | Complete | 142 (52.0) | 131 (48.0) | 1 | | 1 | |
| | Incomplete | 23 (32.4) | 48 (67.6) | 1.41 (1.15-1.73) | **0.001** | 1.57 (1.25-1.98) | **<0.001** |
| **Household livelihood** | Crop production | 84 (41.0) | 121 (59.0) | 1 | | 1 | |
| | Government officer | 8 (53.3) | 7 (46.7) | 0.79 (0.45-1.38) | 0.406 | 0.72 (0.44-1.17) | 0.186 |
| | Housewife | 4 (57.1) | 3 (42.9) | 0.72 (0.31-1.72) | 0.468 | 0.88 (0.36-2.14) | 0.778 |
| | Hunting | 6 (22.2) | 21 (77.8) | 1.32 (1.04-1.66) | **0.020** | 1.17 (0.89-1.55) | 0.254 |
| | Market (food and animal trade) | 13 (61.9) | 8 (38.1) | 0.65 (0.37-1.13) | **0.124** | 0.60 (0.36-0.99) | **0.046** |
| | No occupation/ unemployed | 30 (83.3) | 6 (16.7) | 0.28 (0.13-0.59) | **0.001** | 0.27 (0.12-0.57) | **0.001** |
| | Self employed | 14 (66.7) | 7 (33.3) | 0.56 (0.30-1.05) | 0.069 | 0.58 (0.32-1.05) | 0.070 |
| | Student | 6 (50.0) | 6 (50.0) | 0.85 (0.48-1.51) | 0.574 | 0.69 (0.38-1.26) | 0.229 |
| **Tried measures to keep out bats** | No | 56 (45.2) | 68 (54.8) | 1 | | | |
| | Yes | 109 (49.5) | 111 (50.5) | 0.92 (0.75-1.13) | 0.430 | | |
| **Cover water** | No | 144 (48.6) | 152 (51.4) | 1 | | | |
| | Yes | 21 (43,8) | 27 (56.3) | 1.10 (0.83-1.44) | 0.514 | | |
| **Have storage room for food** | No | 151 (50.7) | 147 (49.3) | 1 | | 1 | |
| | Yes | 14 (30.4) | 32 (69.6) | 1.40 (1.12-1.76) | 0.003 | 1.26 (0.84-1.87) | 0.266 |

*(Continued)*

**Table 3.** (Continued)

| Variable | Attribute | Exposure to bats | | Crude PR (95% CI) | P-values | Adjusted PR (95% CI) | P-values |
|---|---|---|---|---|---|---|---|
| | | Low (n = 165) | High (n = 179) | | | | |
| Commonly visits caves | Boys | 58 (47.5) | 64 (52.5) | 1 | | | |
| | Girls | 5 (71.4) | 2 (28.6) | 0.54 (0.17-1.78) | 0.315 | | |
| | Men | 94 (46.5) | 108 (53.5) | 1.02 (0.82-1.26) | 0.861 | | |
| | Women | 8 (61.5) | 5 (38.5) | 0.73 (0.36-1.49) | 0.391 | | |
| Direct exposure to bats | Bat Bites | 2 (40.0) | 3 (60.0) | 1 | | | |
| | Bat Scratch | 2 (22.2) | 7 (77.8) | 1.30 (0.58-2.88) | 0.524 | | |
| | Collection of dead bats | 161 (50.5) | 158 (49.5) | 0.83 (0.40-1.70) | 0.604 | | |
| | Eating Bat Meat | 0 (0.0) | 10 (100.0) | 1.67(0.81-3.41) | **0.162** | | |
| | Slaughtering of bats | 0 (0.0) | 1 (100.0) | 1.67(0.81-3.41) | **0.162** | | |
| Aware of diseases spread by bats | No | 154 (49.2) | 159 (50.8) | 1 | | | |
| | Yes | 11 (35.5) | 20 (64.5) | 1.27 (0.96-1.69) | 0.998 | | |

During the FGDs, some of the respondents mentioned that they are aware of some of the diseases that are spread by bats such as Ebola, Marburg and COVID 19.

*"We hear that when you eat bat meat, you might get Ebola, Marburg and COVID 19, of recent, we have had these diseases kill so many people in a short time and can easily spread from one person to another in our area"*-**FGD, Ntandi**

Some of the respondents were aware of the measures put in place to reduce the number of bats in their households. In one of the FGDs in Ntandi Town Council, it was highlighted that eating bats is one of the ways of reducing their numbers in the communities.

*"Some of the measures that can reduce bats in our communities include eating them as food, using sticks to beat them so that they can run away."* –**FGD, Ntandi**

Destruction of crops was mentioned by the respondents as one of the dangers of bats. It was further investigated in one of the women groups using one of the participatory epidemiology methods specifically proportional piling and it clearly showed that bats destroy fruits and mostly the guavas as shown in Fig 2 below.

In response to crop destruction by bats, the study revealed during one of the KIIs that bats are consumed as an alternative to crops that have been destroyed by bats as indicated below.

*"Here in Harugale, bats destroy our crops that we feed on thus leaving us with little food, which is not enough for us, we also go and hunt for them and eat them because of what they did to our food. It is revenge"*-**One of the participants in an FGD in Harugale**

Almost two thirds of the respondents 207/344 (60.2%) said that they are disturbed by the presence of bats in their household because of their smell, bat guano, extra cleaning die to littering of bat guano disease spread and the noise 97/207 (46.6%), 19/207 (9.1%), 11/207 (5.3%), 71/207 (34.1%) 10/207 (4.8%) respectively. Almost all the respondents who were disturbed by presence of bat guano were from Burondo 10/11 (90.1%). Majority of the respondents who were not comfortable with their children being exposed to bats were from Ntandi 79/203 (38.9%). Generally, most of the respondents said that they do not feel comfortable living with bats 295/344 (85.8%). Almost three quarters of the respondents believed that

**Table 4. Awareness on human-bat interaction among communities in Bundibugyo district.**

| Variable | Category | Burondo n (%) | Harugale n (%) | Ntandi n (%) | Total n (%) |
|---|---|---|---|---|---|
| **Aware of bat entry points** | Through the door | 1 (8.3) | 11 (91.7) | 0 (0.0) | 12 (3.5) |
| | Through the roof | 31 (19.7) | 47 (29.9) | 79 (50.3) | 157 (46.5) |
| | Through the ventilator | 7 (38.9) | 1 (5.6) | 10 (55.6) | 18 (5.2) |
| | Through the window | 8 (32.0) | 17 (68.0) | 0 (0.0) | 25 (7.3) |
| **Aware of who usually visits caves** | Boys | 32 (26.2) | 58 (47.5) | 32 (26.2) | 122 (35.5) |
| | Girls | 7 (0.0) | 0 (0.0) | 0 (0.0) | 7 (2.0) |
| | Men | 64 (31.7) | 41 (20.3) | 97 (48.0) | 202 (58.7) |
| | Women | 10 976.9) | 3 (21.3) | 0 (0.0) | 13 (3.8) |
| **Aware of bats in schools** | No | 14 (51.9) | 13 (48.1) | 0 (0.0) | 27 (7.8) |
| | Yes | 99 (31.2) | 89 (28.1) | 129 (40.7) | 317 (92.2) |
| **Aware of bats in places of worship** | No | 74 (50.0) | 83 (31.2) | 109 (41.0) | 266 (77.3) |
| | Yes | 39 (50.0) | 19 (24.4) | 20 (25.6) | 78 (22.7) |
| **Aware of diseases spread by bats.** | No | 89 (28.4) | 96 (30.7) | 128 (40.9) | 313 (91.0) |
| | Yes | 24 (77.4) | 6 (19.4) | 1 (3.2) | 31 (9.0) |
| **Aware of trade in bat meat** | No | 92 (30.0) | 96 (31.3) | 119 (38.8) | 307 (89.2) |
| | Yes | 21 (56.8) | 6 (16.2) | 10 (27.0) | 37 (10.8) |
| **Aware of bat dangers** | Carry disease pathogens, e.g., Marburg | 25 (21.4) | 25 (21.4) | 67 (57.3) | 118 (34.3) |
| | Destruction of crops | 47 (35.9) | 29 (22.1) | 55 (42.0) | 131 (38.1) |
| | Their bites or scratches are harmful | 4 (50.0) | 1 (12.5) | 3 (37.5) | 8 (2.3) |
| | Collapsing of buildings and ceilings | 13 (81.3) | 1 (6.3) | 2 (12.5) | 16 (4.7) |
| | Bad smell | 24 (33.3) | 46 (63.9) | 2 (2.8) | 72 (20.9) |
| **Type of information heard/seen about bats** | Cultural beliefs about bats | 14 (38.9) | 22 (61.1) | 0 (0.0) | 36 (10.5) |
| | Dangers of bats | 26 (35.1) | 15 (20.3) | 33 (44.6) | 74 (21.5) |
| | Diseases from bat-human interaction | 32 (26.2) | 26 (21.3) | 64 (52.5) | 122 (35.5) |
| | Reduction of bat-human interaction | 39 (45.9) | 14 (16.5) | 32 (37.6) | 85 (24.7) |
| | Importance of bats | 2 (7.4) | 25 (92.6) | 0 (0.0) | 27 (7.8) |
| **Source of information on bats** | Radio | 27 (27.3) | 40 (40.4) | 32 (32.3) | 99 (28.8) |
| | Friends | 6 (100.0) | 0 (0.0) | 0 (0.0) | 6 (1.7) |
| | Health workers/VHTs | 30 (38.0) | 26 (32.9) | 23 (29.1) | 79 (23.0) |
| | Newspaper | 4 (9.3) | 7 (16.3) | 32 (74.4) | 43 (12.5) |
| | Social media | 8 (34.8) | 15 (65.2) | 0 (0.0) | 23 (6.7) |
| | Television | 38 (40.8) | 14 (15.0) | 42 (44.3) | 94 (27.3) |
| **Most trusted information source** | Radio | 38 (33.9) | 18 (16.1) | 57 (50.0) | 113 (33.9) |
| | Friends | 2 (100.0) | 0 (0.0) | 0 (0.0) | 2 (0.6) |
| | Health workers/VHTs | 30 (34.1) | 26 (29.5) | 32 (36.4) | 88 (25.6) |
| | Newspaper | 4 (8.0) | 13 (26.0) | 33 (66.0) | 50 (14.5) |
| | Television | 39 (42.9) | 45 (49.5) | 7 (7.7) | 91 (26.5) |
| **Information put out in understood language** | No | 54 (42.2) | 51 (39.8) | 23 (18.0) | 128 (37.2) |
| | Yes | 59 (27.3) | 51 (26.3) | 106 (49.1) | 216 (62.8) |
| **Clarity of the information about bats** | No | 72 (28.8) | 57 (22.8) | 121 (48.4) | 250 (72.7) |
| | Yes | 41 (46.3) | 45 (47.9) | 8 (8.5) | 94 (27.3) |
| **Need for more information** | No | 13 (65.0) | 7 (35.0) | 0 (0.0) | 20 (5.2) |
| | Yes | 100 (30.9) | 95 (29.3) | 129 (39.8) | 324 (94.2) |
| **Type of information needed.** | Cultural beliefs about bats | 5 (22.7) | 17 (77.3) | 0 (0.0) | 22 (6.4) |
| | Dangers of bats | 26 (70.3) | 11 (29.7) | 0 (0.0) | 37 (10.8) |
| | Reduction of bat-human interaction | 73 (26.4) | 74 (26.8) | 129 (46.7) | 276 (80.2) |
| | Importance of bats | 9 (0.0) | 0 (0.0) | 0 (0.0) | 9 (2.6) |

bats contaminate food 256/344 (74.4%). More than two thirds 254/344 (73.9%) of the respondents strongly expressed their worry for bats presence in places of worship. More than half of the respondents 203/344 (59%) confirmed that they feel it is not safe for children to be around bats as shown in Table 5 below.

**Beliefs towards bats among communities in Bundibugyo district**

There were several community beliefs highlighted by community members during the KIIs and FGDs that were related to human-bat interaction and associated with high or low exposure to bats. Communities attributed human-bat interaction especially consumption of bats to several beliefs which included but not limited to treatment of diseases, increasing life span, increasing sexual stamina, love portions, sources of wealth, strengthening relationships among others. On the other hand, bats were believed to be misfortunes and signs of bad luck and some of the beliefs included bats being used for rituals, miscarriages, bats being drivers of night dancers by picking peoples' hair. Consumption of bat meat leading to mis-carriages was the most mentioned belief and bats presence in the house being associated with accumulating wealth was the least mentioned belief during the key informants and FGDs as shown below as in Table 6 below.

In one of the KIIs, it was highlighted that consumption of bat meat increases the life span of people and the raw blood of bats cures anemia after being drunk by a young anemic child as shown below.

*"These big bats, we usually call them "**emirima**" are very delicious we go out to the farms with our bows and arrows and stones to kill these big bats, they are as sweet as chicken, when you have killed it, you first boil it in water, then you deep fry it and there you realize that it is the sweetest meat available. We even give our children the raw blood of bats as it is believed to cure anemia. It also increases our lifespan when we eat these bats."* **KII-Batwa Community in Burondo**

During one of the FGDs with a women group in Harugale, it was highlighted that bats are considered evil, and a bad omen once consumed by women.

*"It is believed that when a pregnant woman eats a bat, she is very likely to give birth to a child whose face looks like that of a bat, who will cry like a bat and be restless like a bat. Therefore, we women do not like it at all but when our*

**Table 5. Perceptions towards human-bat interaction.**

| Variable | Category | Burondo n (%) | Harugale n (%) | Ntandi n (%) | Total n (%) |
|---|---|---|---|---|---|
| **I am disturbed by bats 'presence** | No | 11 (22.4) | 12 (13.6) | 26 (53.1) | 49 (14.2) |
| | Yes | 49 (23.7) | 78 (37.7) | 80 (38.6) | 207 (60.2) |
| **I feel disturbed by bat smell** | Yes | 36 (37.1) | 39 (40.2) | 22 (22.7) | 97 (46.6) |
| **I feel disturbed by bat guano** | Yes | 1 (5.3) | 13 (68.4) | 5 (26.3) | 19 (9.1) |
| **I am disturbed by guano cleaning due to bats** | Yes | 10 (90.9) | 1 (9.1) | 0 (0.0) | 11 (5.3) |
| **I am afraid of bats ability to spread diseases** | Yes | 3 (4.2) | 24 (33.8) | 44 (62.0) | 71 (34.1) |
| **I am disturbed by bat noise** | Yes | 0 (0.0) | 1 (10.0) | 9 (90.0) | 10 (4.8) |
| **It is safe for children to be around bats** | No | 48 (26.3) | 76 (38.9) | 79 (38.9) | 203 (59.0) |
| | Yes | 8 (24.2) | 11 (33.3) | 14 (42.4) | 33 (9.6) |
| **I think bats can contaminate food** | No | 41 (46.6) | 45 (51.1) | 2 (2.3) | 88 (25.6) |
| | Yes | 72 (28.1) | 57 (22.3) | 127 (49.6) | 256 (74.4) |
| **I feel comfortable living with bats** | No | 102 (34.6) | 90 (30.5) | 103 (34.9) | 295 (85.8) |
| | Yes | 11 (22.4) | 12 (24.5) | 26 (51.3) | 49 (14.2) |
| **I am worried by bats presence in worship places** | No | 27 (30.0%) | 33 (36.7) | 30 (33.3) | 90 (26.2) |
| | Yes | 86 (33.9) | 69 (27.2) | 99 (39.0) | 254 (73.8) |

**Table 6. Beliefs towards bats among communities.**

| Beliefs on bats | FGDs No (3 per subcounty) | | | | | | | | | KII No | | | TOTAL/12 |
|---|---|---|---|---|---|---|---|---|---|---|---|---|---|
| | 1 | 2 | 3 | 4 | 5 | 6 | 7 | 8 | 9 | 1 | 2 | 3 | |
| Bats are sent by night dancers | 1 | 0 | 0 | 1 | 0 | 1 | 0 | 0 | 1 | 0 | 0 | 1 | 5 |
| Eating bats increases lifespan | 1 | 1 | 0 | 1 | 0 | 1 | 1 | 0 | 1 | 0 | 0 | 1 | 7 |
| Drinking bat blood cures anemia in children | 0 | 0 | 1 | 0 | 1 | 0 | 1 | 0 | 1 | 1 | 0 | 0 | 5 |
| Bats are used for witchcraft | 0 | 1 | 0 | 1 | 1 | 0 | 1 | 0 | 1 | 0 | 1 | 1 | 7 |
| Mothers give birth to children looking like bats when they eat bats | 1 | 0 | 1 | 0 | 1 | 1 | 1 | 0 | 0 | 0 | 0 | 1 | 6 |
| Bat consumption can cause miscarriages | 0 | 1 | 1 | 0 | 1 | 1 | 1 | 0 | 1 | 0 | 1 | 1 | 8 |
| Bats in houses are a source of wealth | 1 | 0 | 0 | 0 | 0 | 0 | 0 | 0 | 1 | 0 | 1 | 0 | 3 |
| Eating bats increases sexual activity/stamina | 1 | 0 | 0 | 0 | 0 | 0 | 1 | 1 | 0 | 0 | 1 | 0 | 4 |
| Bat meat can be used as a love portion | 1 | 0 | 0 | 0 | 1 | 0 | 0 | 0 | 0 | 1 | 1 | 0 | 4 |

*husbands bring them at home, we prepare it for them, but we cannot eat it, we don't want our babies to be like bats."*
**Women group FGD in Harugale**

## Discussion

Human-bat interaction is increasingly being recognized to trigger zoonotic outbreaks that highly affect humans. However, few research studies have revealed the nature of specific interactions that occur between bats and humans leading to pathogen spillover. Studies on bats have been predominantly scientific with little concern for the social perspective leaving a huge data gap. Humans have interacted with bats for ages [18]. Yet, as human activities impact the ecosystem globally, it becomes very important to understand social and cultural factors associated with human-bat synchronicity. Such knowledge and awareness can be the foundation for developing feasible, effective and culturally suitable local conservation interventions which are valuable to both humans and bats [19]. Following the recent Ebola Virus Disease in Uganda and the Corona Virus Disease Outbreak in January 2023 and March 2020 [20] respectively, exposure to zoonotic disease risks at the human-bat interface needs to be assessed, understood and documented.

Sociodemographic factors as well as livelihood activities are essential for consideration in prevention of zoonotic disease exposure at the human-bat interface and developing sustainable programs. In previous studies on the human-bat interaction, sociodemographic and socioeconomic variables such as livelihood activities have an implication on the discrepancy in their interaction with bats, knowledge, awareness as well as the perceptions towards human-bat interaction [21]. A previously conducted study about on human exposure to wild animals in the Sankuru Province of the Democratic Republic of the Congo revealed that the level of exposure of humans to bats was significantly affected by demographic factors such as sex, residential status, sub county, age, gender, and level of education [22]. These findings are similar and were observed among the responses of the respondents in the Burondo, Harugale and Ntandi sub counties where the level of exposure of humans to bats was significantly associated with sex of the household head, livelihood activity, residential status, and religion. Similarly, a recent study conducted in Bundibugyo district also indicated that sex, occupation, area setting were significantly associated with bat exposure among persons living close to bat roosts [23].

Furthermore, the current study revealed that families headed by men are significantly associated with high exposure to bats compared to the families headed by females, in addition, men and boys visit the caves more regularly compared to the women and girls. This is because they are fearful, more adventurous and more interested in hunting wildlife species for food, medicine and fun compared to their counterparts who express more sympathy and love to bats [24]. It was noted that majority of the households in Bundibugyo district were depending on crop farming such as cocoa and vanilla which

have fluctuating seasons. A study conducted in Phillipines suggests that seasonal crops triggers hunger and poverty among community thus, they resort to hunting and trading of bats as source of food and income [17].

According to the study, the households occupied by the Batwa were significantly associated with high exposure to bats. This was attributed to being a primitive hunting and gathering group of people living in isolation and not aware of the current laws in place regarding interaction with wildlife because of their cultural beliefs and past practices. These findings are consistent with a study conducted on knowledge, perceptions and exposure to bats in Bundibugyo district which revealed that hunters were 10 times more likely to get exposed to bats compared to other livelihood activities [23]. The Batwa are entirely reliant on hunting wildlife species for their food, medicine, rituals, and other important activities. This contradicts with the Uganda Wildlife Act, 2019 which aims to provide for the conservation and sustainable management of wildlife, strengthen wildlife conservation and management [25].

The study also revealed that high bat exposure is significant in urban areas compared to rural areas. A study about bats in urban areas conducted in Brazil revealed that bats can adapt to urban landscapes, their ecological flexibility and capability to disperse makes them resilient to urban areas. They take advantage and are able to roost in the urban forest fragments, fruit trees, street lamps, schools, health facilities, places of worship and other buildings in the urban areas [26]. This is also consistent with a previous study which confirms that bats have the ability to occupy people's houses given their capacity to acclimatize to human altered landscapes where buildings and other structures are positioned (e.g., roofs, concrete pipes, and bridges) [27]. This allows bats to adapt to urban landscapes thus their ability to survive in human residences when they lose their habitat. This characteristic enabled bats to be present and plentiful in the sub counties that were studied [28].

From the current study, high bat exposure was more significant in incomplete structures compared to the complete structure. Studies have however been conducted and suggested various house construction preferences that reduce bat entry into the houses. These measures may be helpful in reducing and mitigating the risk for human exposure to zoonotic disease transmission at the human-bat interface. A recent study conducted in Taita-Taveta county in Kenya commended that the modification and proper sealing of structures, particularly in modern tall, cement-walled constructions, may decrease bat presence [29]. Continent-wide purpose to similarly modify housing characteristics based on bat selection parameters may be key in decreasing bat habitation in buildings [30]. Although our current study mentioned radios as the most trusted sources of information, previous studies have highlighted that increasing information sharing on the communities most trusted and available media channels through educational discussions, such as sub county-specific discussions speaking about community mitigation needs, cultural beliefs about bats, dangers of bats, how to reduce bat-human interaction and the importance of bats in the ecosystem are key to curtailing human-bat interaction in communities thus reducing the risk of pathogen exposure and extent of contact with bats [31].

Our current study highlighted several beliefs related to bats some of which would either increase or reduce the rate of human exposure to bats. Communities usually hate and fear bats and consider them as terrific or mythical creatures linked with calamity given their physical make up and relationship with century-old myths [17]. In other countries, bats are thought to be blind rats that have wings and are carrying rabies, leading to blindness, extracting human blood and plucking human hair [32].The various ethnicities in Bundibugyo district during the study has several cultural beliefs that were related to human-bat interaction and that greatly influenced how humans interact with bats thus increasing or decreasing their risk of zoonotic disease transmission at the human-bat interface. Similarly, Malagasy culture in Madagascar has vast cultural taboos and ancestral customs [33]. Some myths, such as the belief that women who are pregnant should not eat bats because they will either have miscarriages or their babies will be looking like them and behaving like them, are not only exclusive to Madagascar. Likewise, the same belief about regarding pregnant women is also in Ghana [34]. Studies indicating that there are beliefs that talk about bats being associated with wealth, increase in sexual activity and increase in lifespan were not available as indicated in the current study.

## Conclusion

The study revealed 52% high exposure to bats among human communities in Bundibugyo district. This was associated with households being headed by males, households with in the Batwa communities residing in urban areas, trading of food and animal products, no occupation and residing in incomplete houses. There was high exposure to bats in Ntandi compared to Burondo and Harugale. Bat numbers increase during the rainy season and decrease during the dry season. Bats were also observed in schools and places of worship, and these had the highest number of individuals interacting with bat excreta and thus increased spillover risk. Although the results are not descriptive of the entire country, they avail valuable information for commencing interventions geared towards increasing awareness among high-risk populations regarding the potential risk of bat exposures and how to live safely with bats.

## Supporting information

**S1 Appendix. Questionnaire.**
(PDF)

**S2 Appendix. Key informant interview guide.**
(PDF)

**S3 Appendix. Focus group discussion guide.**
(PDF)

**S4 Appendix. Informed consent form.**
(PDF)

**S5 Appendix. Minimal dataset.**
(XLSX)
**Community perceptions towards bats**

## Acknowledgments

We would like to thank the community members of Bundibugyo District, the District Officials (The Chief Administrative Officer, the District Health Officer, The District Veterinary Officer and the District Health Educator) for the support rendered to us during the study. We also thank the subcounty chiefs who guided the research teams into the respective households to conduct the interviews.

## Author contributions

**Conceptualization:** James Natweta Baguma, Shamilah Namusisi, Lesley Rose Ninsiima, Clovice Kankya, Charles Drago Kato.

**Data curation:** James Natweta Baguma, Rogers Musiitwa, Douglas Bulafu, Charles Drago Kato.

**Formal analysis:** James Natweta Baguma, Bridget Nagawa Tamale, Mathius Amperiize, Douglas Bulafu.

**Funding acquisition:** James Natweta Baguma.

**Investigation:** James Natweta Baguma, Shamilah Namusisi, Clovice Kankya.

**Methodology:** James Natweta Baguma, Lesley Rose Ninsiima, Bridget Nagawa Tamale, Mathius Amperiize, Charles Drago Kato.

**Project administration:** James Natweta Baguma, Shamilah Namusisi, Clovice Kankya, Charles Drago Kato.

**Resources:** James Natweta Baguma, Shamilah Namusisi, Rogers Musiitwa, Douglas Bulafu, James Muleme.

 

**Software:** James Natweta Baguma, Lesley Rose Ninsiima, Douglas Bulafu.

**Supervision:** James Muleme, David Musoke, Clovice Kankya, Charles Drago Kato.

**Validation:** James Natweta Baguma, James Muleme, David Musoke, Charles Drago Kato.

**Visualization:** James Natweta Baguma, Rogers Musiitwa.

**Writing – original draft:** James Natweta Baguma, Clovice Kankya.

**Writing – review & editing:** James Natweta Baguma.

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
