## [Decision Letter · Decision Letter 0]

14 May 2025

PGPH-D-24-02910

Bat-Human Interactions and associated factors among communities in Bundibugyo District, Uganda: A Cross-sectional Study

Dear Dr. James Natweta Baguma,

Thank you for submitting your manuscript to PLOS Global Public Health. After careful consideration, we feel that it has merit but does not fully meet PLOS Global Public Health’s publication criteria as it currently stands. Therefore, we invite you to submit a revised version of the manuscript that addresses the points raised during the review process.

Please consider addressing comments from reviewer 1 and reviewer 2. The authors are also requested to prepare the response file by showing the revised changes directly in the response file, instead of only providing line numbers. This will facilitate the review of the changes made in the manuscript. In case of changes of multple pages, line numbers may be mentioned only.

We look forward to receiving your revised manuscript.

Kind regards,

Dr. Rebeca Sultana

Academic Editor

Journal Requirements:

1. We have amended your Competing Interest statement to comply with journal style. We kindly ask that you double check the statement and let us know if anything is incorrect. 2. Your current Financial Disclosure states, “This study was supported by the Strategies to Prevent Spillover Project”. However, your funding information on the submission form indicates that you received funding from “USAID”. Please indicate by return email the full and correct funding information for your study and confirm the order in which funding contributions should appear. Please be sure to indicate whether the funders played any role in the study design, data collection and analysis, decision to publish, or preparation of the manuscript. 3. Please provide separate figure files in .tif or .eps format.For more information about figure files please see our guidelines:  https://journals.plos.org/globalpublichealth/s/figures https://journals.plos.org/globalpublichealth/s/figures#loc-file-requirements 4. Some material included in your submission may be copyrighted. According to PLOS’s copyright policy, authors who use figures or other material (e.g., graphics, clipart, maps) from another author or copyright holder must demonstrate or obtain permission to publish this material under the Creative Commons Attribution 4.0 International (CC BY 4.0) License used by PLOS journals. Please closely review the details of PLOS’s copyright requirements here: PLOS Licenses and Copyright. If you need to request permissions from a copyright holder, you may use PLOS's Copyright Content Permission form. Please respond directly to this email or email the journal office and provide any known details concerning your material's license terms and permissions required for reuse, even if you have not yet obtained copyright permissions or are unsure of your material's copyright compatibility.  Potential Copyright Issues:a. Figure 1: please (a) provide a direct link to the base layer of the map (i.e., the country or region border shape) and ensure this is also included in the figure legend; and (b) provide a link to the terms of use / license information for the base layer image or shapefile. We cannot publish proprietary or copyrighted maps (e.g. Google Maps, Mapquest) and the terms of use for your map base layer must be compatible with our CC-BY 4.0 license.  Note: if you created the map in a software program like R or ArcGIS, please locate and indicate the source of the basemap shapefile onto which data has been plotted. If your map was obtained from a copyrighted source please amend the figure so that the base map used is from an openly available source. Alternatively, please provide explicit written permission from the copyright holder granting you the right to publish the material under our CC-BY 4.0 license. Please note that the following CC BY licenses are compatible with PLOS license: CC BY 4.0, CC BY 2.0 and CC BY 3.0, meanwhile such licenses as CC BY-ND 3.0 and others are not compatible due to additional restrictions.  If you are unsure whether you can use a map or not, please do reach out and we will be able to help you. The following websites are good examples of where you can source open access or public domain maps: * U.S. Geological Survey (USGS) - All maps are in the public domain. (http://www.usgs.gov) * PlaniGlobe - All maps are published under a Creative Commons license so please cite “PlaniGlobe, http://www.planiglobe.com, CC BY 2.0” in the image credit after the caption. (http://www.planiglobe.com/?lang=enl) * Natural Earth - All maps are public domain. (http://www.naturalearthdata.com/about/terms-of-use/) 5. We have noticed that you have uploaded Supporting Information files, but you have not included a list of legends. Please add a full list of legends for your Supporting Information files after the references list.

Additional Editor Comments (if provided):

Reviewers' comments:

Reviewer's Responses to Questions

**Comments to the Author**

1. Does this manuscript meet PLOS Global Public Health’s publication criteria ? Is the manuscript technically sound, and do the data support the conclusions? The manuscript must describe methodologically and ethically rigorous research with conclusions that are appropriately drawn based on the data presented.

Reviewer #1: Yes

Reviewer #2: Yes

2. Has the statistical analysis been performed appropriately and rigorously?

Reviewer #1: No

Reviewer #2: Yes

3. Have the authors made all data underlying the findings in their manuscript fully available (please refer to the Data Availability Statement at the start of the manuscript PDF file)?

Reviewer #1: Yes

Reviewer #2: No

4. Is the manuscript presented in an intelligible fashion and written in standard English?

Reviewer #1: Yes

Reviewer #2: Yes

5. Review Comments to the Author

Reviewer #1: General Comments

This study aims to address an important gap in literature that pertains to the socio-demographic drivers of human-animal interfaces which modulate the risk of zoonotic spillovers. The main rationale of the study was to perform survey through semi-structured interviews targeting community households and key informants. While the study has made some progress towards gaining insight in the sociological factors influencing human-bat interactions, there are several weaknesses to the study design. Firstly, the process of selecting socio-demographic and socio-economic features/ variables is not clearly motivated. Secondly, the study could benefit from further statistical analyses which show which of the factors drive/ influence disease risk. An example of such analysis would be the increased or decreased risk of Ebola incidence in the geographical area during outbreak year. Thirdly, there is no clear explanation of which of the results come from Key Informants and which come from selected household representatives in the focus group discussion. A clear explanation for convergence and divergence of opinions backed by solid framework is lacking. Finally, the insights gained in this study have not been employed sufficiently to inform further research and hence lack sufficient research impact. The generalizability of the outcomes, as authors themselves mention, is poor and wouldn’t apply to the whole country or group of neighbouring countries.

The introduction doesn’t specify why a particular district is targeted for this study. Is it because of high zoonotic disease incidence or is it due to increase in urban population of bats or human migration? What about practices common to this region, does that affect the increased risk of zoonotic spillover?

Tables and Figures do not have numbers, Table 1, Table 2…hard to see where they are referenced in the text

No line numbers after page 21

Abstract/ Keywords

34: Semi-structured?

35: No need to mention the software in abstract if it is already mentioned in Methods section

39, 40: Same, no need to mention stata or Atlas

40: What do you mean our study was limited by recall bias? How does it add to abstract?

43: What is APR? Where is it’s full name explained?

49: Focus Group Discussions (Unnecessary capitalization)

57: remove bats as keyword, does not add to the search optimization

Introduction

82: What do you mean by “firmly established”? Which emergent diseases come from bats? I don’t think the authors mean that all newly emergent diseases come from bats? The explanation is insufficient.

101-102: Why is interaction likely common in Bundibugyo district? Provide some context here.

Materials and Methods

120: UBOS abbreviation

126: Highlight?

130-131: What is facilitated?

136: How are household heads defined?

139: Sometimes the authors use abbreviations and sometimes full terms like focus group discussions

142: Explain how proportional piling was performed. Reference?

145: Highlight

154-155: Also 146, add as reference and access date rather than hyperlink

155-156: Needs to be mentioned in Introduction

164: (1965; 15)

Results

240-260: Repeated information from Table: Socio-demographic factors… Additionally, no Table 1 or Table 2.

297 and elsewhere: What is CPR and APR?

Figure Proportional piling…: What are the numbers within parentheses?

Table Factors associated with…: What do Crude PR and Adjusted PR signify? Are they on log scale? Are those with 1 reference levels?

Table Beliefs towards…: What are the scores here? Medians? Rating by respondents? Number of respondents? Why is total divided by 12? Is it an average or a total?

Discussion

human-bat synchronicity: What do the authors mean by synchronicity?

A recent study conducted in Taita-Taveta county in Kenya commends: Recommends?

Communities usually hate and fear bats…: Why are beliefs relevant here? Did the authors explore relationship between religion and these beliefs? Where are those results?

Conclusion

52% of what? Communities, counties, households, respondents?

References

11: Formatting

13: Access date incomplete

21: Formatting

24: Publishing town/ country not mentioned, edition not mentioned

Reviewer #2: In this manuscript, Baguma et al investigate human-bat interactions in Bundibugyo District, Uganda, and identify factors associated with exposure to bats. The study provides valuable insights into the dynamics of human-bat interactions. The topic is not novel, but it highlights the need for intervention to reduce exposure to bats and mitigate zoonotic disease transmission; there are several areas that require improvement.

Major Concerns

1. Authors state at lines 104-107, “However, information on bat hunting and consumption practices, as well as indirect contacts, is scarce. Understanding of the risk factors, knowledge, awareness, and perceptions of communities towards bat‐related diseases is also limited, without showing what is known in order to better understand between available data in the same area and their study.

2. What findings are classified as the drivers of human-bat interaction? risk of human exposure? awareness of bats? mitigation measures, profiled dangers and/or benefits of bats, and perceptions. The manuscript could benefit from a more detailed and organized presentation of results.

3. We have in the result section, lines 286-287, we have this sentence: “Respondents during one of the key informant interviews highlighted that human-bat interaction varies with the seasons when bats are present and absent. “ The method section and questions did not cover the seasons. Can authors explain it in the Materials and Methods section?

4. The discussion section could be expanded to provide more context on the implications of findings and potential interventions to reduce human-bat interactions.

Minors concerns

Abstract:

The abstract is well constructed and provides the necessary and complete information about the study findings. Please define APR.

Introduction:

The reasons for Bat-Human interactions are not clearly defined, and should be described, addressing the possible and known possibilities to date.

Methods:

Line 119: remove the second full stop “district..”

Lines 130-131: “These are facilitated by the two rainy seasons of March-May and July-November. “ Farmers (of coffee, vanilla, and cocoa) and what else, other activity is facilitated by the two rainy seasons?

Lines 195-203 are these questions used for statistical analysis?

Who administered the questionnaire in this study?

Results:

Whenever possible, proportions and percentages should be accompanied by the actual numerator and denominator from which they were derived.

Lines 264-267, is it a result? If we consider at result, how many of the included participants state that?

Please add a number for each table and add it to the text

6. PLOS authors have the option to publish the peer review history of their article (what does this mean? ). If published, this will include your full peer review and any attached files.

**Do you want your identity to be public for this peer review?** For information about this choice, including consent withdrawal, please see our Privacy Policy .

Reviewer #1: **Yes: ** Pranav Shrikant Kulkarni

Reviewer #2: No

---

## [Editor Report · Decision Letter 1]

15 Jul 2025

PGPH-D-24-02910R1

Bat-Human Interactions and associated factors among communities in Bundibugyo District, Uganda: A Cross-sectional Study

Dear Dr. Katie Laird,

Thank you for submitting your manuscript to PLOS Global Public Health. After careful consideration, we feel that it has merit but does not fully meet PLOS Global Public Health’s publication criteria as it currently stands. Therefore, we invite you to submit a revised version of the manuscript that addresses the points raised during the review process.

Please address the comments provided by the reviewers in a separate document and send back to us for final decision. 

We look forward to receiving your revised manuscript.

Kind regards,

Rebeca Sultana

Academic Editor
---

## [Editor Report · Decision Letter 2]

29 Jul 2025

Bat-Human Interactions and associated factors among communities in Bundibugyo District, Uganda: A Cross-sectional Study

PGPH-D-24-02910R2

Dear Authors,

We are pleased to inform you that your manuscript 'Bat-Human Interactions and associated factors among communities in Bundibugyo District, Uganda: A Cross-sectional Study' has been provisionally accepted for publication in PLOS Global Public Health.

Best regards,

Rebeca Sultana

Academic Editor